# ROSE: Reduced Overhead Stereo Event-Intensity Depth Estimation

## Abstract

Stereo depth estimation using event cameras is a promising approach for real-time vision tasks, offering low-latency, high-speed data capture. However, existing methods often suffer from high computational overhead, limiting their real-time applicability. To address these challenges, we introduce ROSE (Reduced Overhead Stereo Event and Intensity) a Real-Time framework for efficient depth estimation from events and intensity images. Current approaches rely on dense networks that fail to scale with increasing data complexity, constraining both accuracy and speed. In contrast, ROSE incorporates lightweight event representation networks and optimizes the stereo matching process to reduce model size and computational load without compromising accuracy. We replace conventional network components with efficient spatio-temporal representations and streamline adaptive aggregation modules, reducing computational complexity by 1000× compared to previous methods. Furthermore, we adapt event grouping strategies to better align with intensity images, improving the quality of depth estimation under various lighting and motion conditions. Extensive experiments on the DSEC and MVSEC benchmarks demonstrate that ROSE achieves real-time performance, boosting frame rates to 32.2 FPS on DSEC and 66.9 FPS on MVSEC while maintaining competitive depth accuracy. This marks a significant improvement over prior work in terms of speed and scalability, making ROSE a viable solution for real-time stereo depth estimation in resource-constrained environments. Our code and models will be released to support further advancements in the field.

## 1 Introduction

Stereo depth estimation has long been a key challenge in computer vision, traditionally addressed through epipolar geometry and pixel matching across intensity image frames (RGB or grayscale) (Mayer et al., 2016; Laga et al., 2020; Smolyanskiy et al., 2018; Poggi et al., 2022) have gained prominence, with state-of-the-art (SoTA) approaches involving feature extraction, cost aggregation, depth estimation, and refinement (Zhang et al., 2019; Xu & Zhang, 2020; Xu et al., 2023). Despite these advancements (Laga et al., 2020), key challenges remain: $(i)$ depth inaccuracies due to lighting conditions (Jeon et al., 2016; Sharma & Cheong, 2018; Heo et al., 2013), $(ii)$ motion blur from conventional camera limitations (Xu & Jia, 2010), and $(iii)$ high computational costs for processing high-resolution intensity images.

Recent research has explored integrating event cameras to address these challenges (Gallego et al., 2020; Zheng et al., 2023; de Queiroz Mendes et al., 2021; Jia et al., 2022; Xu & Zhang, 2020; Rancon et al., 2022). While intensity cameras capture detailed scene information, they suffer from motion blur, latency, and poor performance under challenging textures and lighting conditions (Mueggler et al., 2014; Lagorce et al., 2015; Gallego et al., 2018). In contrast, event cameras (neuromorphic cameras) (Lichtsteiner et al., 2006; 2008) capture pixel-level changes asynchronously, offering advantages like low latency and a wide dynamic range. These qualities make event cameras less susceptible to motion blur and lighting distortions. However, their sparse, asynchronous data outputs pose considerable challenges for integration with intensity-based depth estimation methods.

Event cameras generate asynchronous, sparse event streams represented by pixel coordinates $(x, y)$, polarity, and timestamps. Due to the vast number of events, stacking event data is a common method for dense depth estimation (Mostafavi I. et al., 2020; 2021; Nam et al., 2022; Zhang et al., 2022).

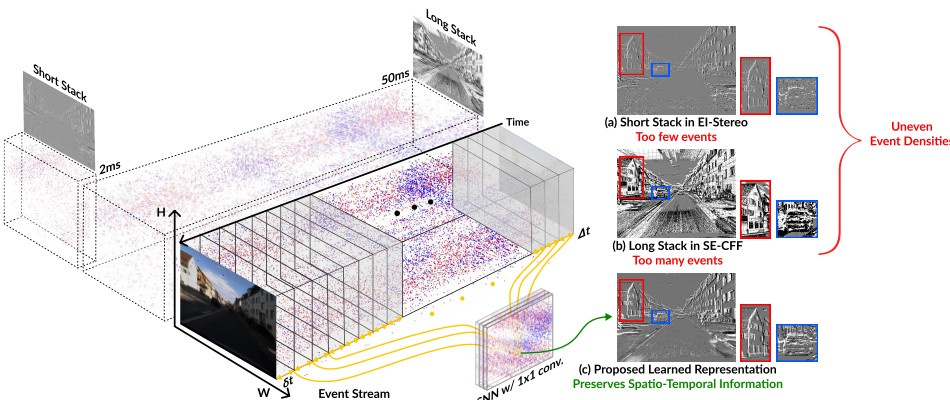

Figure 1: **Comparing the Event Representations learned by SE-CFF, EI-Stereo, and ROSE.**
Short Stack, *i.e.*, stacking a small number of events, contains little structural information of objects.
Long Stack, *i.e.*, stacking a large number of events, causes motion blur and loss in object structure
and texture. Our spiking neuron-based event representation aims to preserve the structural informa-
tion by using temporal information of event stream. (Section 3.2).

Table 1: **Architectural analysis of recent event-intensity depth estimation frameworks**. We
showcase the event representations, total parameter count, and the inference speed measured in
frames per second (FPS). ROSE architecture is designed to bypass the use of resource-intensive
components that typically cause bottlenecks in event representation.

|  | EI-Stereo | SE-CFF | SCSNet | SMC-Net | **ROSE (Proposed)** |
|---|---|---|---|---|---|
| Event Representation | SBT Stack | Concentrated Stack | Voxelized Event | Event Queue | **ROSE Image** |
| FPS | 10.0 (MVSEC) | 9.3 (DSEC) | 10.1 (DSEC) | 8.3 (MVSEC) | **32.2** (DSEC) / **66.9** (MVSEC) |
| # Parameters | - | 10.0 M | 17.4 M | - | **3.6 M** (DSEC), **1.0 M** (MVSEC) |

**\*Citations**: EI-Stereo (Mostafavi et al., 2021b); SE-CFF (Nam et al., 2022); SCSNet (Cho & Yoon, 2022a); SMC-Net (Cho & Yoon, 2022b); SBT
Stack (Mostafavi et al., 2021a); Event Queue (Tulyakov et al., 2019)

However, simplistic stacking approaches can result in information loss. Short stacks, based on time
or event number (Wang et al., 2019; Mostafavi et al., 2021a), may fail to capture enough visual
information, while long stacks may exacerbate motion blur and structural loss, as demonstrated
in Figure 1. Recent research has sought to mitigate these issues by improving event representation
methods to preserve structural integrity.

As shown in Table 1, SoTA methods like SE-CFF (Nam et al., 2022) and EI-Stereo (Mostafavi
et al., 2021b) use complex networks and event stacking techniques to handle motion blur and pre-
serve structural information. These methods are resource-intensive and slow. SE-CFF, for example,
operates at only 9 FPS on the DSEC (Gehrig et al., 2021b) dataset, limiting the real-time appli-
cability. This underscores the necessity for computationally efficient solutions that maintain depth
accuracy.

To address these limitations, we propose **ROSE** (*Reduced Overhead Stereo Event and Intensity
Network*). It includes a novel event representation model in addition to further network elements
designed to achieve SoTA depth estimation with real-time performance. The core of ROSE is a
spiking neuron-based event representation learning that is robust to light artifacts and motion blur,
enabling it to outperform prior methods in both speed and efficiency.

Compared to SE-CFF (Nam et al., 2022), ROSE runs at over $3\times$ faster at 32.2 FPS while requiring
$64\%$ fewer parameters. On MVSEC (Zhu et al., 2018), ROSE achieves 66.9 FPS, far surpassing the
8.3 FPS of SMC-Net (Cho & Yoon, 2022b) with $96\%$ fewer parameters than EIT (Ahmed et al.,
2021), making it a highly efficient and desirable solution for real-time applications.

**Contributions of this work** are summarized as follows:

- **Event Stream Analysis**: We thoroughly analyze event streams from the DSEC and MVSEC
  datasets, systematically examining their characteristics to identify an optimal set of events for
  enhanced integration with intensity images, thereby improving depth estimation.

- **Event Representation**: We present an innovative event representation model utilizing spiking neurons, substantially improving robustness against light artifacts and motion blur. This model also reduces latency and enhances overall performance.
- **Computational Efficiency**: We remove computationally expensive network components to improve real-time performance with substantially fewer parameters compared to previous arts.
- **Unmatched Speed and SoTA Performance on DSEC**: ROSE attains an exceptional inference speed of 32.2 FPS alongside state-of-the-art depth estimation metrics.
- **Real-Time Inference with Superior or Comparable Performance on MVSEC**: ROSE achieves 66.9 FPS, ($8\times$ faster than similar models) with sustained or enhanced depth metrics.

## 2 RELATED WORK

### 2.1 EVENT CAMERAS IN STEREO DEPTH ESTIMATION

As introduced in Section 1, event cameras *stream* asynchronously in tuples relative to pixel locations, diverging from the conventional concept of a 'frame'. This asynchronous streaming is challenging in general for independent representation and also for alignment with frames from other modalities. To tackle the aforementioned challenges, prior works attempt to align event data with conventional deep learning methodologies, by processing it either in a stacked format (Wang et al., 2019) or as discrete event volumes (Zhu et al., 2019). When combined with image data, these approaches have demonstrated improved performance over the uni-modal image-only setting (Mostafavi et al., 2021b; Nam et al., 2022; Cho & Yoon, 2022a;b; Gehrig et al., 2021a).

EI-Stereo (Mostafavi et al., 2021b) uses a Recycling Network that aims to reconstruct details of RGB images and event stacks. SCSNet (Cho & Yoon, 2022a) uses cascaded Neighbor Cross Similarity Feature modules to align event and image data. SE-CFF (Nam et al., 2022) proposes a Concentration Network that learns to stack 10 event frames, where a frame is obtained by stacking millions of events. However, a critical aspect often overlooked by prior methods is the effective utilization of *temporal information* inherent to event data. Hence, unlike prior works, our work focuses on learning representations that preserve the *spatial and temporal information* present in the event data.

### 2.2 IMPORTANCE OF EVENT-STREAM SPLITTING

When learning event representations, we split the event data stream by a time interval of size "$\delta t$" for easier computations. This is because the event stream contains millions of data points which makes processing the individual events difficult, leading to out-of-memory errors. Selecting an optimal value of $\delta t$ is crucial as it heavily governs the quality of the input event data. As presented in Figure 1, grouping excessive events (large $\delta t$), will lead to *motion blur* and not including enough events (small $\delta t$), increases the risk of missing details and preventing effective learning. Chosing the correct $\delta t$ is challenging as the optimal value can be different for each dataset as it depends on the properties of the event stream and the scene being captured.

There have been efforts to circumvent picking an optimal value for $\delta t$. For instance, SE-CFF (Nam et al., 2022) propose "Concentration Network" that arbitrarily splits the event stream into 10 different groups of various $\delta t$ values, irrespective of the dataset. While they achieve SoTA results on DSEC, such arbitrary splitting leads to groups of non-uniform event densities, thus leading to some containing too many events and some containing too few events. On the other hand, EI-Stereo (Mostafavi et al., 2021b) propose a "Recycling Network" (based on *e2sri* (Mostafavi I. et al., 2020; 2021)) that uses 3 event groupings obtained by splitting the stream by very small $\delta t$ time-intervals. But, these groupings contain too little event information, thus requiring multiple cycle iterations in the Recycling Network, leading to excessive computations at low FPS rates. The effects of performing such arbitrary splitting can directly be observed in their learned event representations and predicted depth maps, where they contain many depth inaccuracies due to motion blur and light artifacts. Samples are portrayed in Figure 4 and Figure 5 for reference.

We hypothesize that splitting the event stream by arbitrary $\delta t$ values makes it harder to learn good representations, worsening motion blur. Adding to that, such arbitrary values lead to vastly different results depending on the dataset, since each dataset has highly different event streams. Hence, these challenges necessitate a systematic way of selecting an optimal event grouping strategy.

Building on top of these observations, we perform an in-depth analysis where the aim is to use a fixed value for $\delta t$ to group the event stream per dataset. Through our analysis, we find and propose an optimal way to select $\delta t$ for the most desirable stereo depth performances, outlined in Section 3.1. We follow this up by proposing techniques that can considerably improve the efficiency and FPS rates while achieving SoTA performances on the popular DSEC dataset (Gehrig et al., 2021b).

# 3 TECHNIQUES FOR REAL-TIME EFFICIENT DEPTH ESTIMATION

## 3.1 ANALYZING AND OPTIMALLY SPLITTING THE EVENT STREAM

As mentioned in Section 2.2, we aim to find an optimal way to select the value of $\delta t$ that reduces the effects of motion blur and artifacts. We scrutinize and analyze the statistics of the intensity and event stream groupings to lend us a hand in picking the optimal $\delta t$ value. We take a careful look at two properties of the event groupings: ($i$) *standard deviations* $\sigma$, and ($ii$) *event densities* $\rho$.

We first split the DSEC event stream by $\delta t$, where $\delta t \in \{2\text{ms}, 10\text{ms}, 25\text{ms}, 50\text{ms}\}$, and obtain multiple different event groupings. This allows us to calculate each grouping's $\sigma$ value. We also take a close look at the intensity images and calculate the $\sigma$ values of its 3-channeled RGB images to be $[0.249, 0.258, 0.266]$. We report these numbers in Table 2. Next, we move on to analyzing and calculating the event groupings' $\rho$ values and report them in Table 3. We can notice right away for DSEC that ($a$) the standard deviation of the event groupings obtained by $\delta t = 2\text{ms}$ is the closest to the RGB intensity images' standard deviations, and ($b$) they contain $40K$ event data points on average. These two factors are crucial in making accurate disparity maps without suffering from adverse motion blur. By analyzing MVSEC similarly, we observe that $\delta t = 10\text{ms}$ is the optimum.

To back our analysis, ablation on the effects of $\delta t$ can be found in Section 4.3.1. The results support our hypothesis in Section 2.2 as well, that large and arbitrary values of $\delta t$ would lead to worse motion blur and depth estimation metrics. Qualitative results in Figure 7 further support this claim.

Table 2: **Standard Deviations** of the event stream grouped by $\delta t \in \{2\text{ms}, 10\text{ms}, 25\text{ms}, 50\text{ms}\}$, for DSEC (Gehrig et al., 2021b) and MVSEC (Zhu et al., 2018). The selected option is the most appropriate for $\delta t$ because: (1) it is close to the dataset's intensity standard deviation, and (2) it ensures an ample quantity of event data when compared with the event densities in Table 3.

| Dataset | Intensity Images | $\delta t = 2\text{ms}$ | $\delta t = 10\text{ms}$ | $\delta t = 25\text{ms}$ | $\delta t = 50\text{ms}$ |
|---------|------------------|--------|---------|---------|---------|
| DSEC | $[0.249, 0.258, 0.266]$ | **0.306** | 0.802 | 1.345 | 1.928 |
| MVSEC | 0.168 | 0.109 | **0.248** | 0.387 | 0.548 |

Table 3: **Event Densities** of DSEC and MVSEC data at $\delta t \in \{2\text{ms}, 10\text{ms}, 25\text{ms}, 50\text{ms}\}$.

| Dataset | $\delta t = 2\text{ms}$ | $\delta t = 10\text{ms}$ | $\delta t = 25\text{ms}$ | $\delta t = 50\text{ms}$ |
|---------|--------|---------|---------|---------|
| DSEC | **40K** | 200K | 500K | 1M |
| MVSEC | 0.6K | **3K** | 7.5K | 15K |

## 3.2 LEARNING SPATIO-TEMPORAL EVENT REPRESENTATIONS EFFICIENTLY

Convolutional layers use receptive fields which act as an inductive bias that helps in learning meaningful patterns present in the neighboring pixels (Krizhevsky et al., 2012). The extent to which neighboring pixels are utilized is dependent on the kernel size, and increasing this size leads to an increase in parameters. SE-CFF's Concentration Net (Nam et al., 2022) is computationally expensive because it implements multiple convolutional layers of varying kernel sizes, making it expensive and inefficient. EI-Stereo's Recycling Network (Mostafavi et al., 2021b) suffers from this as well, since the inputs and outputs and re-iterated multiple times, thus requiring even more computations.

To address this problem, we propose utilizing just two convolutional layers, each with a kernel of size $1 \times 1$. To facilitate better temporal learning, we use the Leaky Integrate-and-Fire (LIF) spiking neurons (Hunsberger & Eliasmith, 2015). Spiking neurons have shown great promise (Cordone et al., 2021), as they naturally handle asynchronous data streams (Rathi et al., 2023). By using an LIF neuron model. we can effectively capture the dynamics of asynchronous event data by modeling the membrane potential changes.

The core motivation behind using spiking neurons and a kernel of size $1 \times 1$ is that we want to focus on the cross-channel (temporal) correlations at the pixel level, without any influence from its neighbors. Following these changes, our proposed event representation network requires significantly fewer FLOPs and energy, as seen in Table 4. We see that our event representation network is extremely resource-friendly, requiring only a $\frac{1}{1000}$-th of Nam et al. (2022)'s FLOPs.

Next, to objectively gauge the proposed event representation network's efficiency compared to SE-CFF, we replace the Concentration Network with our proposed network in SE-CFF's pipeline and retrain the network. We keep everything else the same, including implementing the same Stereo Matching Network as SE-CFF. Note that EI-Stereo and SE-CFF use the same Stereo Matching Network, and the only thing different is the event representation network.[1] We report the results in Table 5. Although SE-CFF's metrics are currently still better, we see an improvement in computational complexity (FLOPs) for our method as expected.

Lastly, to gauge our event representation network's efficiency with other stereo-matching backbones, we perform ablation studies where we replace the traditional CNN-based Stereo Matching Network with two Transformer-based backbones, STTR and Lightweight STTR (Li et al., 2021). The results can be found in Section 4.3.2.

Table 4: **Energy and computational efficiency on the DSEC dataset**. ROSE demands $1000\times$ fewer FLOPs and consumes $65000\times$ less energy, all while delivering FPS rates exceeding $3\times$ ($\delta t = 2$ms) those of (Nam et al., 2022).

| Method | Data Resolution | FLOPs ($\downarrow$) | Energy (mJ, $\downarrow$) | VRAM (GB, $\downarrow$) |
|---|---|---|---|---|
| Concentration Network (Nam et al., 2022) | $640 \times 480$ | 1770 G | 4094.0 | - |
| Proposed Event Representation Network | $640 \times 480$ | **1.72 G** | **0.063053** | **0.86** |

Table 5: **DSEC results and efficiency of learning representations through our proposed event representation network.** We nearly match SE-CFF's performances with $1000\times$ fewer FLOPs using our proposed event representation network. Note that for fairness, *we implement the same Stereo Matching Network as SE-CFF (Nam et al., 2022)*.

| Method | FLOPs ($\downarrow$) | MAE ($\downarrow$) | 1PE ($\downarrow$) | 2PE ($\downarrow$) | RMSE ($\downarrow$) |
|---|---|---|---|---|---|
| Concentration Network (Nam et al., 2022) | 1770 G | 0.364 | 4.844 | 0.840 | 0.818 |
| Proposed Event Representation Network | **1.72 G** | 0.370 | 4.923 | 0.930 | 0.854 |

## 3.3 REDUCING MODEL SIZE

We achieve higher FPS rates and lower computational complexity by implementing the methods from Section 3.1 to 3.2. While this significantly enhances the model we still need further considerations to reach real-time inference.

Therefore, in order to improve results and make depth estimation possible in real-time, we aim to reduce the overall size of our Stereo Matching Network by using fewer adaptive aggregation modules (3 layers instead of 6). Adaptive aggregation modules are computationally heavy as they contain multiple nested for-loops that prove to be time-wise and space-wise complex. Therefore, halving the number of these modules would have an improvement in the throughput and the number of FLOPs. We name this, *"Enhanced SMNet"*. In addition to the Enhanced SMNet, we make a few changes to the learning rate scheduler as well. We outline the differences in the Appendix A. With our proposed changes, we for the first time surpass the performances of SE-CFF across the board. Furthermore, we achieve this at an unprecedented real-time FPS and significantly fewer FLOPs.

Also, to objectively gauge our proposed event representation network's improvements compared to EI-Stereo's "Recycling Network", we directly replace it with our proposed representation network while keeping everything else the same and report it in Table 6. In other words, we follow EI-

---

[1]EI-Stereo (Mostafavi et al., 2021b) uses Recycling Network to learn event representations, whereas SE-CFF (Nam et al., 2022) uses Concentration Network. The Stereo Matching Network in both of them are the exact same architectures, based on the AANet (Xu & Zhang, 2020).

Stereo's pipeline but swap the Recycling Network with our proposed representation network. Note that EI-Stereo and SE-CFF use the same Stereo Matching Network (refer to Section 3.2).

Table 6: Comparison of Enhanced Stereo Matching Network (SMNet) with the original SMNet on DSEC. ROSE achieves superior results with significantly fewer (50%) FLOPs. We further include the pipelines from previous arts for comprehensive analyses.

| Method | MAE ($\downarrow$) | 1PE ($\downarrow$) | 2PE ($\downarrow$) | RMSE ($\downarrow$) | FPS ($\uparrow$) | Total FLOPs ($\downarrow$) |
|---|---|---|---|---|---|---|
| *Following EI-Stereo (Mostafavi et al., 2021b)'s Pipeline* | | | | | | |
| Recycling Network (Mostafavi et al., 2021b) | 0.396 | 5.814 | 1.055 | 0.905 | - | $\gg$ 3.94 T |
| Proposed Event Representation Network | 0.391 | 5.564 | 1.020 | 0.904 | 23.1 | 1.72 G + 2.17 T $\simeq$ 2.17 T |
| *Following SE-CFF (Nam et al., 2022)'s Pipeline* | | | | | | |
| Concentration Network (Nam et al., 2022) | 0.364 | 4.844 | 0.840 | 0.818 | 9.3 | 1.17 T + 2.17 T = 3.94 T |
| Proposed Event Representation Network | 0.370 | 4.923 | 0.930 | 0.854 | 15.1 | 3.06 G + 2.17 T $\simeq$ 2.17 T |
| *Following Changes in Sections 3.1 to 3.3* | | | | | | |
| **Proposed Event Representation Network** | **0.356** | **4.717** | **0.814** | **0.815** | **32.2** | **1.72 G + 2.01 T $\simeq$ 2.01 T** |

Table 7: **Utilizing Skip Connections to Preserve Spatial Information in Events-only Modality.** ROSE achieves superior or similar results in nearly all metrics at much higher FPS rates. Following Appendix A.4, skip connections are implemented to preserve spatial information in the Events-only setting. **Best** is bolded, underlined follow-up is underlined.

| Method | Modality | FPS ($\uparrow$) | MAE ($\downarrow$) | 1PE ($\downarrow$) | 2PE ($\downarrow$) | RMSE ($\downarrow$) | # Params. ($\downarrow$) |
|---|---|---|---|---|---|---|---|
| EI-Stereo (Mostafavi et al., 2021b) | Events-only | - | 0.529 | 9.958 | 2.645 | 1.222 | - |
| SE-CFF (Nam et al., 2022) | Events-only | 11.3 | 0.519 | 9.583 | 2.620 | 1.231 | 6.9 M |
| DTC-SPADE (Zhang et al., 2022) | Events-only | - | 0.526 | 9.270 | **2.405** | 1.285 | - |
| ROSE with Skip Connections (Proposed) | Events-only | **30.7** | **0.503** | **9.110** | 2.444 | **1.183** | **3.6 M** |

### 3.4 PRESERVING SPATIAL INFORMATION WITH SKIP CONNECTIONS

Until now, we have emphasized the importance of learning the spatio-temporal information present in the event data. When both Intensity and Event data (*i.e.* multimodal) are used for the stereo depth estimation, there is enough spatio-temporal information to facilitate effective learning. On the other hand, using just unimodal Event data is more challenging due to the lack of spatial information, which leads to inferior performance metrics.

To address this disparity in performance, we propose a simple yet effective solution to preserve the spatial information. We introduce *skip connections* between the event data and the event representation network's output. This straightforward solution drastically improves results, leading to near-SoTA results with much higher FPS than prior works on DSEC in the event-only setting, as shown in Table 7. The results when skip connections are not used are reported in Appendix Table 14.

## 4 PUTTING IT TOGETHER

### 4.1 ARCHITECTURE PIPELINE

Our overall pipeline is illustrated in Figure 2, and other than our proposed event stream pre-processing stage, the rest is similar to the overall pipelines of SE-CFF (Nam et al., 2022) and EI-Stereo (Mostafavi et al., 2021b). First, the event data streams and intensity images are supplied by two event cameras. These event streams are then split into $T$ groups each based on a time interval of size $\delta t$, as mentioned in Section 2.2. Next, these $T$ event groups are subsequently passed to our proposed spiking neuron-based event representation network, which generates a 1-channeled *Representation Image*. As there are two cameras, we obtain two Representation Images.

We then concatenate the intensity image with the corresponding Representation Image to obtain a 4-channeled tensor for DSEC. [2] Finally, this tensor is passed into our Enhanced SMNet where

---

[2]For MVSEC, it is a 2-channeled tensor since its intensity image is a 1-channeled grayscale image.

the features are extracted to form two cost volumes. Deformable aggregations and refinements are performed on these cost volumes to produce a multi-level disparity pyramid whose last layer is used as the estimated disparity map. We implement the Smooth $L_1$ objective function, similar to SE-CFF and EI-Stereo. A detailed explanation of the objective can be found in Appendix A.3.

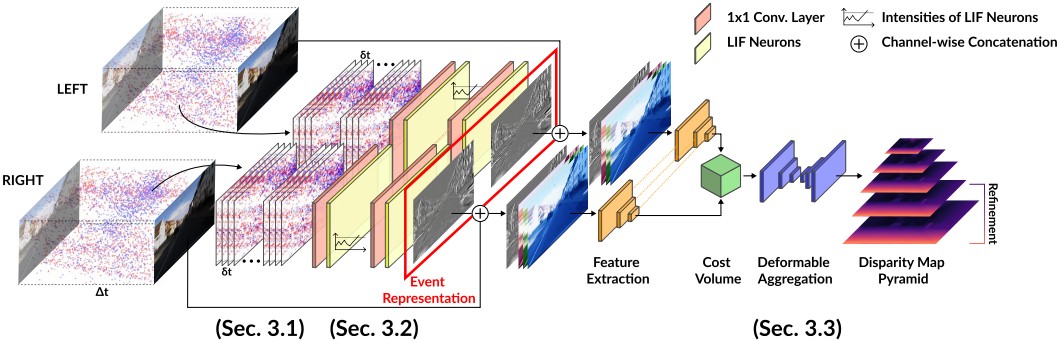

Figure 2: **ROSE's Architecture.** ROSE processes the event frames (Section 3.1) using $1\times1$ convolutions. Events arriving at each pixel get accumulated and encoded by the representation by leveraging LIF neurons (Section 3.2). before finally being used to predict the disparities(Section 3.3).

## 4.2 EXPERIMENTS AND OBSERVATIONS

Official DSEC metrics include Mean Absolute Error, 1-Pixel Error, 2-Pixel Error, and Root Mean Square Error. For MVSEC, the official metrics include Mean Depth Error, Mean Disparity Error, and 1-Pixel Accuracy. Importantly, we also report the inference FPS rates. We report all these metrics in the tables mentioned below. We implement our code using PyTorch (Paszke et al., 2019) and snnTorch (Eshraghian et al., 2023), a library specifically designed for the spiking neurons.

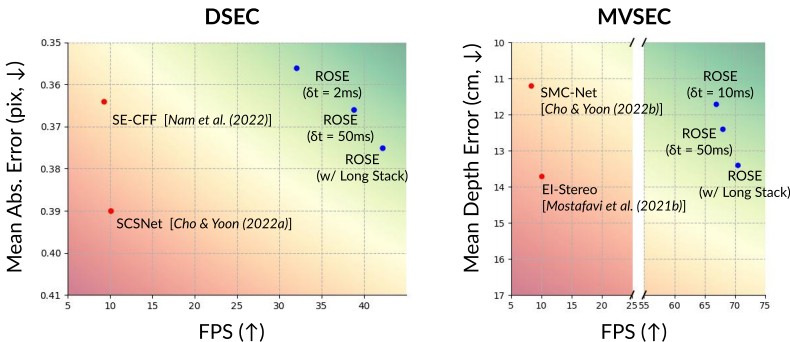

Figure 3: **Comparative Analysis of ROSE and Previous Models.** Benchmarking the inference speed, quantified in frames per second (FPS), against the performance, assessed by mean absolute error (MAE) on DSEC and mean depth error (MDE) on MVSEC, in event-intensity methods on the both datasets. ROSE presents similar or improved MAE (MDE) with vastly upgraded FPS under different $\delta t$ settings (Section 4.3.1). Numbers are derived form the papers and are only presented if both the specific dataset results (MVSEC/DSEC) and FPS values were reported.

Combining all the methodologies in Section 3.1 to 3.3 and Appendix A.4, we train our proposed model on the DSEC (Gehrig et al., 2021b) and MVSEC (Zhu et al., 2018) datasets, and compare its performance with the current SoTA works. We summarize our results in fig. 3 and Table 8. We also compare the computational complexities of the event representation networks and the entire models of the current DSEC SoTA (SE-CFF) with our work. We report these metrics in Tables 5 and 6.

Several key observations can be made from Tables 5, 6, and 8. First, ROSE, *consistently outperforms SE-CFF (Nam et al., 2022) on all metrics with* $64\%$ *fewer parameters* on DSEC. Second, ROSE *performs the SoTA depth estimation at an unprecedented 32.2 FPS.* This is $3\times$ the FPS rate of the current fastest model, SCSNet (Cho & Yoon, 2022a). Third, ROSE achieves $8\times$ the FPS rate of SMC-Net (Cho & Yoon, 2022b) with near-SoTA results on all metrics on MVSEC with require $96\%$

Table 8: **Results for Performance Comparison on DSEC and MVSEC.** Both Event and Intensity data are used. RGB images for DSEC and grayscale images for MVSEC. **Best** is in bold.

| Method | DSEC | | | | MVSEC | | | | | | (DSEC / MVSEC) | |
| | MAE ($\downarrow$) | 1PE ($\downarrow$) | 2PE ($\downarrow$) | RMSE ($\downarrow$) | MDE (cm, $\downarrow$) | | MDisE (pix, $\downarrow$) | | 1PA (%, $\uparrow$) | | FPS ($\uparrow$) | # Params. ($\downarrow$) |
| | | | | | Split1 | Split3 | Split1 | Split3 | Split1 | Split3 | | |
| EI-Stereo (Mostafavi et al., 2021b) | 0.396 | 5.814 | 1.055 | 0.905 | 13.7 | 22.4 | - | - | 89.0 | 88.1 | - / 10.0 | - / - |
| EIT (Ahmed et al., 2021) | - | - | - | - | 14.2 | 19.4 | 0.55 | 0.75 | 92.1 | 89.6 | - / - | - / 22.3 M |
| SE-CFF (Nam et al., 2022) | 0.364 | 4.844 | 0.840 | 0.818 | - | - | - | - | - | - | 9.3 / 18.2 | 10.0 M / - |
| SCSNet (Cho & Yoon, 2022a) | 0.390 | 5.670 | 0.990 | 0.850 | 11.4 | **13.5** | 0.38 | **0.39** | **94.7** | **94.0** | 10.1 / - | 17.4 M / - |
| SMC-Net (Cho & Yoon, 2022b) | - | - | - | - | **11.2** | 14.5 | **0.37** | 0.52 | 94.3 | 92.0 | - / 8.3 | - / - |
| ROSE (Ours) | **0.356** | **4.717** | **0.814** | **0.815** | 11.7 | 15.8 | 0.41 | 0.58 | 93.7 | 90.7 | **32.2 / 66.9** | **3.6 M / 1.0 M** |

fewer parameters compared to EIT (Ahmed et al., 2021). Lastly, we observe in Tables 5 and 6 that our proposed event representation network *is $1000\times$ more computationally efficient with regards to FLOPs*. Compared to SE-CFF's 1770 G FLOPs, ROSE requires only 1.72 G FLOPs on DSEC. These indicate ROSE's ability to be highly efficient, highly performant, and capable of real-time depth estimation. Note that *on MVSEC, ROSE uses only 0.016 G FLOPs*, which is $1000\times$ fewer compared to its FLOPs count on DSEC. We could not compare with previous works' FLOPs counts as no code has been published.

Figure 4 presents an unbiased qualitative comparison between our method and other existing approaches. We emphasize key areas of interest with colored boxes. Depth predictions generated by ROSE exhibit *sharper details* and *more accurate* object shapes, underscoring their superior quality. In contrast, the depth predictions from SE-CFF and EI-Stereo display hazy, light artifacts, and blurred object shapes and edges.

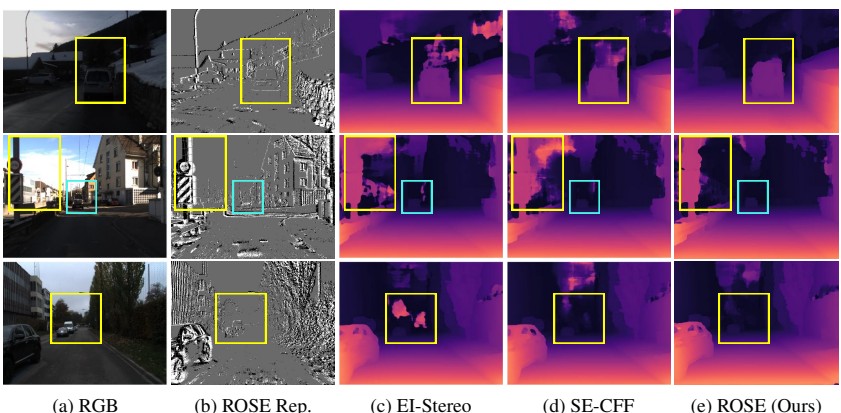

| (a) RGB | (b) ROSE Rep. | (c) EI-Stereo | (d) SE-CFF | (e) ROSE (Ours) |

Figure 4: **Qualitative Comparison on DSEC.** Figure illustrating qualitative results: RGB Images, RoSE Images, and Disparity Predictions of EI-Stereo Mostafavi et al. (2021b), SE-CFF Nam et al. (2022), and our model, ROSE, for 4 different scenes are shown. Predictions of our model contain more detail and have sharper object shapes, while those of Nam et al. (2022) and Mostafavi et al. (2021b) contain less detail and suffer from hazy *'ghosting'* artifacts.

In Figure 5, we can observe that ROSE learns event representations that are more clear and robust to motion blur and hazy artifacts compared to SE-CFF and EI-Stereo. We also demonstrate the adaptability of our approach by displaying the qualitative results on MVSEC in Figure 6. We encourage readers to review our supplementary depth reconstruction video and appendix material.

## 4.3 ABLATION STUDIES

### 4.3.1 SPLITTING EVENT STREAM BY $\delta t$

In Section 3.1, we showed that prior works, specifically SE-CFF and EI-Stereo, pick arbitrary values for $\delta t$ to group by, which worsen motion blur and artifacts. We examined the properties of the event stream and proposed an optimal way to select $\delta t$ that led to robust representations and depth predictions that suffer less from motion blur with improved quality. Table 9 and Figure 7 contain the quantitative and qualitative results, respectively, with a variety of $\delta t$ values on DSEC. These results

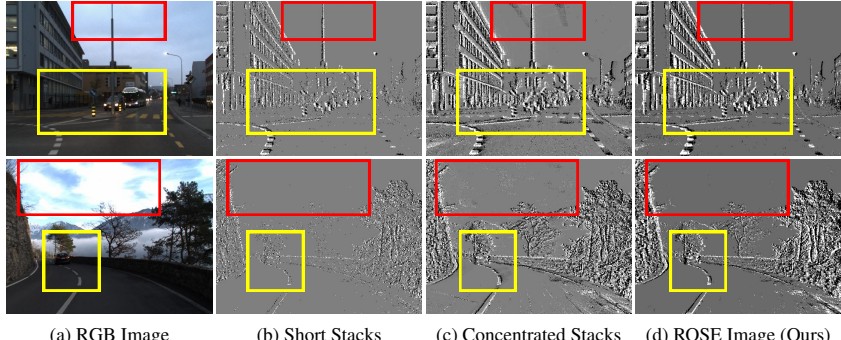

(a) RGB Image     (b) Short Stacks     (c) Concentrated Stacks     (d) ROSE Image (Ours)

Figure 5: **Comparing Event Representations.** (a) contains a scene's corresponding RGB image. (b) *Short Stacks* used by EI-Stereo (Mostafavi et al., 2021b), show loss of structural information. (c) SE-CFF (Nam et al., 2022)'s computationally expensive *Concentrated Stacks* suffer from motion blur. (d) ROSE's Representation Images depict sharp object structures and suffers less from blurs.

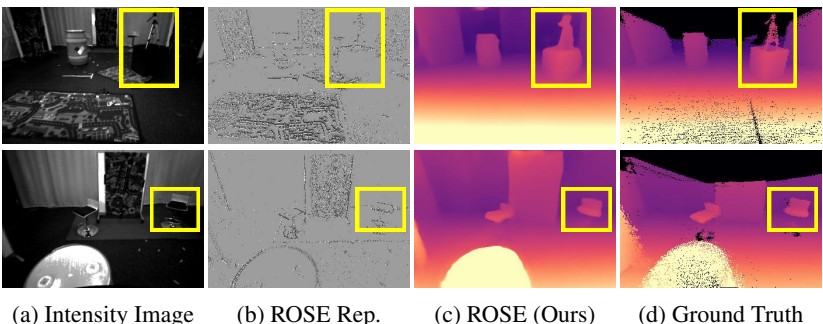

(a) Intensity Image     (b) ROSE Rep.     (c) ROSE (Ours)     (d) Ground Truth

Figure 6: **Qualitative Comparison on MVSEC.** Our model shows sharp and detailed outputs.

back up our claim that arbitrary larger $\delta t$ values (as picked by SE-CFF and EI-Stereo) are bound to suffer more from motion blur and light artifacts.

Table 9: Grouping Events by different $\delta t$ values. 2ms has the best performance as its standard deviation follows intensity images. (50ms skips pre-processing thus the highest FPS.)

| Method | MAE ($\downarrow$) | 1PE ($\downarrow$) | 2PE ($\downarrow$) | RMSE ($\downarrow$) | FPS ($\uparrow$) | FLOPs ($\downarrow$) |
|---|---|---|---|---|---|---|
| SE-CFF (Nam et al., 2022) | 0.364 | 4.844 | 0.840 | 0.818 | 9.3 | 3.94 T |
| ROSE ($\delta t = 50$ms) | 0.367 | 4.871 | 0.881 | 0.847 | **39.4** | **2.02 T** |
| ROSE ($\delta t = 25$ms) | 0.364 | 4.883 | 0.873 | 0.844 | 38.9 | **2.02 T** |
| ROSE ($\delta t = 10$ms) | 0.367 | 4.988 | 0.882 | 0.846 | 38.2 | **2.02 T** |
| **ROSE ($\delta t = $2ms, Proposed)** | **0.356** | **4.717** | **0.814** | **0.815** | 32.2 | **2.02 T** |

### 4.3.2 EFFICIENCY WITH TRANSFORMER-BASED ARCHITECTURES

In order to gauge our proposed event representation network's efficiency in different Stereo Matching architectures, we perform an ablation study with 2 transformer-based architectures: *STTR* and *Lightweight STTR* (Li et al., 2021). We report the FPS and GPU VRAM usage during inference in Table 10. We observe that ROSE is consistently highly efficient for a variety of different image resolutions, reaching up to $14\times$ the FPS rates with just $\frac{1}{7}$-th the VRAM utilization.

### 4.3.3 LEARNING EVENT REPRESENTATIONS WITH LEAKYRELU INSTEAD OF LIF

In Section 3.2, we proposed using Leaky Integrate-and-Fire (LIF) spiking neurons to capture the temporal dynamics of incoming data. We perform ablation studies on the MVSEC dataset in Ta-

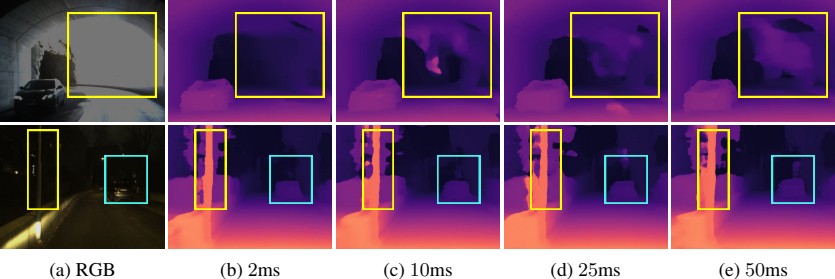

(a) RGB      (b) 2ms      (c) 10ms      (d) 25ms      (e) 50ms

Figure 7: **Qualitative Results for Different $\delta t$ Values.** (a) is RGB images for each scenes; (b), (c), (d), and (e) are predicted disparity maps from scenes. Each column uses a different $\delta t$ value.

Table 10: **Contrasting the GPU memory usage (VRAM) and speed (FPS) at inference time.**

| Method | Data Resolution | VRAM (GB, ↓) | FPS (↑) |
|---|---|---|---|
| STTR (Li et al., 2021) | $640 \times 480$ | 1.80 | 5.77 |
| Lightweight STTR (Li et al., 2021) | $640 \times 480$ | 0.87 | 9.11 |
| **ROSE (Proposed)** | $640 \times 480$ | **0.86** | **32.2** |
| STTR | $960 \times 576$ | 3.80 | 2.76 |
| Lightweight STTR | $960 \times 576$ | 2.00 | 4.42 |
| **ROSE (Proposed)** | $960 \times 576$ | **1.21** | **17.98** |
| STTR | $960 \times 540$ | 7.40 | 1.35 |
| Lightweight STTR | $960 \times 540$ | 2.20 | 4.36 |
| **ROSE (Proposed)** | $960 \times 540$ | **1.07** | **18.93** |

ble 11 and observe that compared to LeakyReLU, our LIF-based event representations lead to improved performances across the board, with lower computational requirements and higher FPS rates.

Table 11: These results verify and back our approach of using the highly efficient LIF-based Event Representation Learning. Although LeakyReLU-based MVSEC (Split 1) metrics are slightly comparable, it demands more FLOPs and energy thus substantially reducing the FPS throughputs.

| Method | FPS (↑) | FLOPs (↓) | MDeE (↓) | MDisE (↓) | 1PA (↓) | Energy (↓) | VRAM (GB) (↓) | # Params. (↓) |
|---|---|---|---|---|---|---|---|---|
| ROSE with LeakyReLU | 55.1 | 17.49 M | 12.4 | 0.43 | 93.2 | 0.031372 | 0.65 | **1.0 M** |
| **ROSE with LIF (Proposed)** | **66.9** | **16.21 M** | **11.7** | **0.41** | **93.7** | **0.000129** | **0.60** | **1.0 M** |

## 5 CONCLUSION

We present ROSE and a novel spiking neuron-based event representation method for stereo depth estimation of event and intensity cameras in real time. We leverage the low-latency nature of event cameras and efficiently process large volumes of event data at high FPS. Our results show that ROSE maintains competitive performance across various time intervals $\delta t$, confirming the importance of preserving temporal information for real-time applications. Additionally, ROSE mitigates common issues associated with event stacking, learning robust event representations with fewer artifacts.

### 5.1 LIMITATIONS AND FUTURE WORK

While ROSE effectively captures temporal information in event data, as discussed in Section 4.3.1, balancing the sparsity of events within $T$ frames remains a key factor. Future work may explore approaches to process events individually. However, the current limitations of computational technology make real-time processing of each event at the camera's temporal resolution a challenge.

## 6 ETHICS STATEMENT

ROSE poses no risk to the general public and does not pose any threats with regard to deepfake images, graphic violence, or offensive material.

## 7 REPRODUCIBILITY STATEMENT

The code will be available at https://github.com/... Moreover, we outline our training procedures in the main text and appendix.

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

## A  APPENDIX

### A.1  DETAILS ABOUT STEREO MATCHING NETWORK

The Feature Extractor extracts the features of the input and produces a list of layers. Each layers' height and width dimensions become progressively smaller while the channel dimension grows larger, resulting in a pyramid-like structure for the output.

The Cost Volume module constructs a *cost volume pyramid* by feature correlation and convolving over the features.

The Adaptive Aggregation Network is based on *Deformable Convolutional Layers* (DC layers). Their advantageous adaptive nature Dai et al. (2017) lead to better results. DC layers also help overcome the implicit dependence on the neighbourhood pixels which is present in traditional convolutional layers.

The Disparity Estimation Pyramid is the disparity estimates for each of the layers in the convolved cost volume pyramid.

The Stereo Disparity Refinement Module is the final part of the stereo-matching network consists of upsampling layers that refine the lower-resolution disparity maps. For DSEC, there are two such layers, whereas there is just one for MVSEC. Post-refinement, the new layers are appended back to the original pyramid to create a $K$-layered disparity pyramid.

### A.2  COMBINING EVENT AND IMAGE MODALITIES

When handling information from different modalities, it is important to combine them effectively to reduce conflicts between them (Gadzicki et al., 2020). As such, there are 2 intuitive approaches to combining them: $(i)$ Input-level (*i.e.*, early-fusion), and $(ii)$ Feature-level (*i.e.*, late-fusion). The former refers to the concatenation of event representations with the images *prior to* feeding them to the feature extractor module (FE). The latter refers to extracting features of each modality using its own individual FE and subsequently concatenating them together. The resulting combined features are then utilized for dense depth prediction.

We evaluate the performance of our proposed architecture in the above two settings in DSEC (Gehrig et al., 2021b), one of the leading benchmarks, and report them in Table 12.

Table 12: Results of *Early-* and *Late-Fusion* of Event and Image Modalities. Performed with $\delta t = $ 2ms on the DSEC benchmark (Gehrig et al., 2021b). **Best** is in bold. Next best is underlined.

| Method | FPS ($\uparrow$) | MAE ($\downarrow$) | 1PE ($\downarrow$) | 2PE ($\downarrow$) | RMSE ($\downarrow$) | # Params. |
|---|---|---|---|---|---|---|
| SE-CFF (Nam et al., 2022) | 9.3 | 0.364 | 4.844 | 0.840 | 0.818 | 10.0 M |
| ROSE *(feature-level)* | 25.3 | 0.365 | 4.827 | 0.871 | 0.845 | 6.5 M |
| ROSE *(input-level)* | **32.2** | **0.356** | **4.717** | **0.814** | **0.815** | **3.6 M** |

Analyzing the results, we see that ROSE with the $(ii)$ feature-level combination approach performs comparable to SE-CFF (Nam et al., 2022) in all metrics (even surpassing it in the *1PE* metric). In the case of $(i)$ input-level combination, ROSE performs better than the other two, with a smaller model and higher FPS. This suggests that input-level early-fusion is a more effective approach to combine the modalities. Overall, thanks to the RoSE module's ability to learn meaningful representations, ROSE performs competitively with real-time capabilities.

Note that $(ii)$ would result in more number of FEs in our case since $(i)$ would only require 1 FE, whereas $(ii)$ requires 2 of them. This results in a considerable increase in ROSE's model size and computation cost which negatively affects its FPS rates. But even in such circumstances, ROSE performs at 25 FPS, which is nearly $3\times$ that of SE-CFF (Nam et al., 2022), while being $35\%$ smaller in size.

## A.3 Details on the Objective Function.

The ground truth, $y \in \mathbb{R}^{H \times W}$, is the disparity map between the left and right images, captured by LiDAR (Gehrig et al., 2021b; Zhu et al., 2018). Given the multi modal inputs to our stereo matching network, $X^L$ and $X^R$, we obtain the $K$-layered pyramid output, $f$.

Let $n \in \{1, 2, ..., K\}$ be the indices of $f$'s layers, where $K$ is the last layer. Then, $f_n$ is the estimated disparity of the $n^{th}$ layer. Prior to calculating the loss, $f_n$ is bi-linearly interpolated to be the same size as $y$, and hence, $f_n \in \mathbb{R}^{H \times W}$.

Therefore, the objective is to minimize the weighted sum of the Smooth $L_1$ losses at each layer of $f$ w.r.t. to the ground truth, $y$:

$$\min \sum_{n=1}^{K} w_n \cdot L\left(y, f_n\right) \tag{1}$$

where $w_n$ is the $n^{th}$ layer's assigned weight. The Smooth $L_1$ loss is preferred because it is less sensitive to outliers and noise compared to $L_2$ loss Barron (2019).

## A.4 Increasing Restart Frequency and Learning Rate

Following the methods from Sections 3.1 to 3.3, we are able to achieve real-time FPS rates with SoTA or near-SoTA results in all metrics. We adopted the "Cosine Annealing with Warm Restarts" scheduler, just like SE-CFF. But in order to surpass their metrics, we propose using a slightly different learning schedule with a larger learning rate.

Unsurprisingly, in our experiments, we noticed that our model can get stuck in local minima as the training progresses. But with "warm restarts" for the learning rate, the model could effectively escape the local minimum and continue the training process.

We hypothesize that increasing the learning rate and providing such warm restarts slightly more often would benefit the model by better escaping the local minimas. Unlike SE-CFF, we set the restart cycle frequency to 50 epochs, and increase the learning rate to be a range of $[5e-6, 1e-3]$ from the original $[5e-8, 5e-4]$. The changes and results are reported in Table 13.

Table 13: **Improving Scheduler for better convergence.** Implemented with the Cosine Annealing with Warm Restarts Schedule. While we already exceed the benchmarks set by (Nam et al., 2022), implementing the proposed scheduler further enhances our performance.

| Method | Max LR | Min LR | MAE ($\downarrow$) | 1PE ($\downarrow$) | 2PE ($\downarrow$) | RMSE ($\downarrow$) | FPS ($\uparrow$) | FLOPs ($\downarrow$) |
|---|---|---|---|---|---|---|---|---|
| CN / Restart every 100 Epochs (Nam et al., 2022) | 5e-4 | 5e-8 | 0.364 | 4.844 | 0.840 | 0.818 | 9.3 | 3.94 T |
| ROSE / Restart every 100 Epochs | 5e-4 | 5e-8 | 0.357 | 4.725 | 0.834 | 0.826 | **32.2** | **2.02 T** |
| ROSE / Restart every 50 Epochs (Proposed) | 1e-3 | 1e-6 | **0.356** | **4.717** | **0.814** | **0.815** | **32.2** | **2.02 T** |

## A.5 Results and Analysis of Event-only Setting

By focusing on the *temporal information* in the low-latency event data, RoSE is able to extract meaningful representations. As a result, ROSE achieves SoTA results with real-time capabilities in the *Event and RGB* multi-modal setting on DSEC (Gehrig et al., 2021b) and MVSEC (Zhu et al., 2018). Now, we aim to analyze and compare its performance in the unimodal *Event-only* setting.

We observed that using *only the event representations* led to suboptimal performances. Since proposed learning mechanism focuses mainly on learning the temporal information, we suspect it might not have learned sufficient spatial relations, thus causing ROSE's poor results. In the multi-modal setting, however, additional spatial information provided through the RGB images can improve learning. We hypothesize that preserving the spatial information could improve ROSE's performances in the unimodal setting.

Recall that we uses the $T$ number of event frames to learn the temporal relations and output the 1-channel representation. Therefore, an intuitive way to preserve spatial relations would be to concatenate the frames along with the learned representation, forming a $(T + 1)$-channel output. This would be essentially similar to *concatenated skip connections* (Huang et al., 2016). We compare the performances of ROSE w/ learned event representation only and w/ also $T$ Frames in Table 14

with existing works (Nam et al., 2022; Zhang et al., 2022; Mostafavi et al., 2021b). Qualitative results are shown in Figure 8. We observe that by implementing our skip connection approach, ROSE achieves SoTA results on MAE, RMSE and 1PE metrics with near-SoTA results on the 2PE metric. ROSE surpasses E-Stereo (Mostafavi et al., 2021b) and SE-CFF (Nam et al., 2022), and performs comparably to DTC-SPADE (Zhang et al., 2022) with few parameters and high FPS rates.

Table 14: **Skip Connection for Preserving Spatial Information.** ROSE achieves SoTA results while operating at much higher FPS rates. Following Section 3.4, we implement a skip connection to preserve the spatial information in the E-only setting. **Best** is in bold. Next best is underlined.

| Method | Modality | FPS (↑) | MAE (↓) | 1PE (↓) | 2PE (↓) | RMSE (↓) | # Params. |
|---|---|---|---|---|---|---|---|
| SE-CFF (Nam et al., 2022) | E | 11.3 | 0.519 | 9.583 | 2.620 | 1.231 | 6.9 M |
| DTC-SPADE (Zhang et al., 2022) | E | - | 0.526 | 9.270 | **2.405** | 1.285 | - |
| ROSE without Skip Connections | E | **31.4** | 0.568 | 11.223 | 3.070 | 1.324 | **3.6 M** |
| ROSE with Skip Connections (Proposed) | E | 30.7 | **0.503** | **9.110** | 2.444 | **1.183** | **3.6 M** |

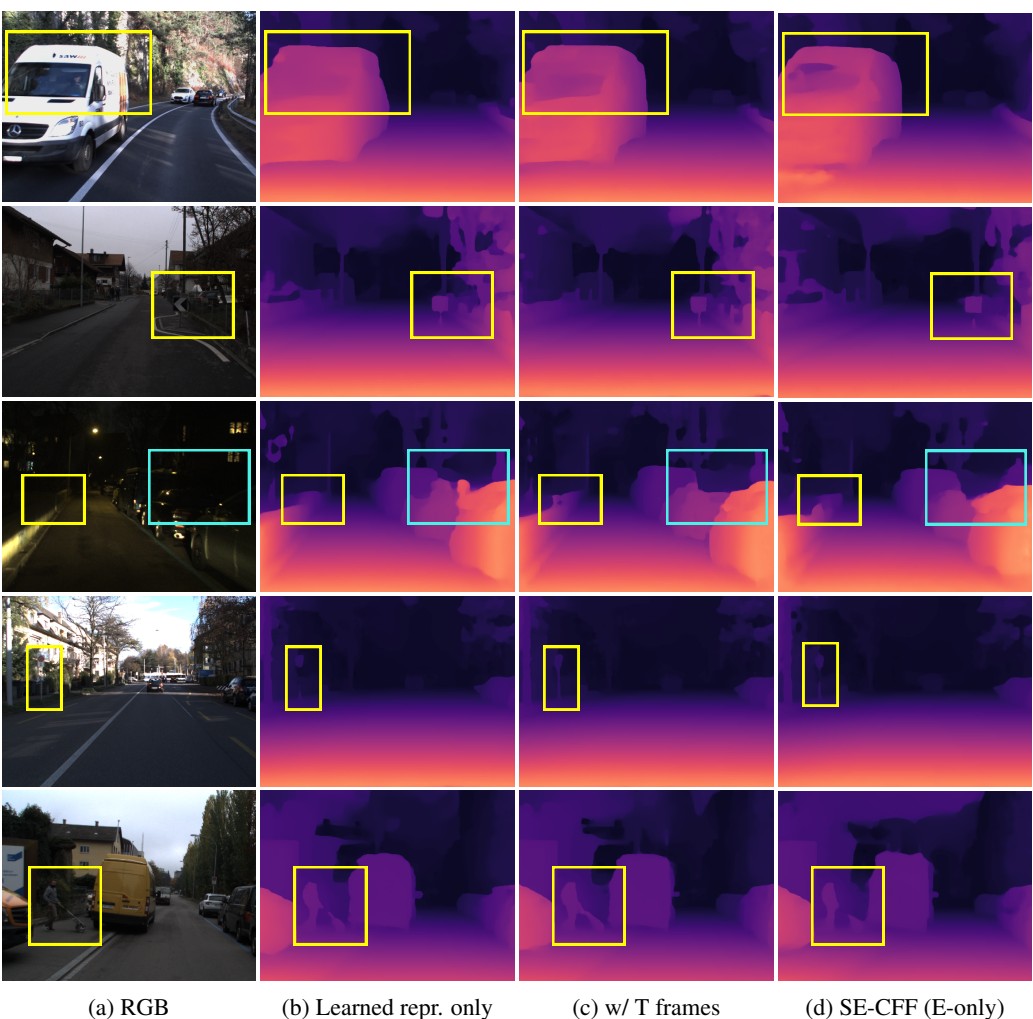

(a) RGB     (b) Learned repr. only     (c) w/ T frames     (d) SE-CFF (E-only)

Figure 8: **Qualitative Comparison on DSEC in Event-only setting.** (a) is RGB image of 5 scenes, (b) and (c) are ROSE's disparity predictions in learned representation only and with T frames settings, respectively. (d) is SE-CFF (Nam et al., 2022)'s disparity predictions in the event-only setting.

### A.6 MORE QUALITATIVE RESULTS

We attached a driving scenario video comparing ROSE and SE-CFF on DSEC. Note that we had to reproduce SE-CFF so the FPS rates displayed in the video may differ from those reported in the paper.

### A.7 DATASETS AND EXPERIMENTAL DETAILS

We used $8 \times$ A6000 GPUs for training and inference.

#### A.7.1 DSEC BENCHMARK.

DSEC (Gehrig et al., 2021b) offers a comprehensive dataset, including event camera data of $480 \times 640$ resolution, high-resolution $1080 \times 1440$ RGB images, and $480 \times 640$ resolution LiDAR disparity maps as ground truth. For optimization, we chose the Adam Kingma & Ba (2015) optimizer. The initial learning rate is set at $5e-4$, with beta values of 0.9 and 0.999, and a weight decay of $1e-4$. To enhance the training process, we employed cosine annealing with warm restarts Loshchilov & Hutter (2017) to schedule the learning rate. Additionally, to address the challenge of depth perception, we set the max disparity to 192. Random cropping is used to make the images $432 \times 576$ resolution, in terms of height and width, respectively. Lastly, for the Disparity Pyramid, we set $K$ to 5, similar to as in StereoDRNet Chabra et al. (2019). Lastly, note that we perform experiments only until $\delta t = 2ms$, because experiments of $\delta t = 1ms$ required more GPU resources than at hand.

#### A.7.2 MVSEC BENCHMARK.

MVSEC (Zhu et al., 2018) includes event data and grayscale images of $260 \times 346$ resolution, generated using the DAVIS sensor. We use the Adam optimizer and setting same as DSEC, but the initial learning rate is set to $1e-3$. We again employed cosine annealing with warm restarts to schedule the learning rate. The max disparity is set to 37 and we use random cropping of resolution $228 \times 312$ on the data. Lastly, we set $K$ to 4.

### A.8 EVENT REPRESENTATION LEARNING WITH OTHER STEREO MATCHING ALGORITHM

We conducted experiments with proposed event representation learning + MobileStereoNet (Shamsafar et al., 2022) to demonstrate the adaptability and versatility of approach as presented in Table 15. This table, with MVSECZhu et al. (2018), further shows how different modalities can be incorporated. And also in Table 15, the multimodal setting performs better than the unimodal setting. Our proposed event representation yields better performance relatively.

Table 15: **MobileStereoNet w/ ours event representation on MVSEC**.

| Method | Modality | MDE (cm, ↓) | MDisE (pix, ↓) | 1PA (%, ↑) |
|---|---|---|---|---|
| Intensity-only | I | 24.95 | 1.62 | 51.42 |
| ROSE image-only ($\delta t = 10ms$) | E | 31.79 | 2.07 | 45.15 |
| ROSE image w/ T frames (T=5) | E | 27.55 | 1.77 | 48.29 |
| Short stack + Intensity ($\delta t = 10ms$) | E+I | 24.59 | 1.58 | 53.69 |
| Long stack + Intensity ($\delta t = 50ms$) | E+I | 25.31 | 1.63 | 51.68 |
| ROSE Image + Intensity ($\delta t = 10ms$) | E+I | **23.63** | **1.52** | **53.76** |

