# OpenReview forum: "ROSE: Reduced Overhead Stereo Event-Intensity Depth Estimation"
_ICLR.cc/2025/Conference — ICLR 2025 Conference Withdrawn Submission_

### Official Review · Reviewer_Eta8 · 2024-11-01

**Soundness:** 2
**Presentation:** 3
**Contribution:** 2
**Rating:** 5
**Confidence:** 3

**Summary:**

This paper tackles the problem of depth estimation using stereo event camera.  It proposes a novel event representation based on a study of the probabilistic characteristics of the events. Furthermore, the widely known Integrate-and-Fire (LIF) spiking neuron model is used to encode the temporal information, and consequently, a computationally efficient representation network is presented. By combining a reduced-size stereo matching network (e.g. AANet) and also considering preserving spatial information through skip connections, the results obtained are comparable to those of SoTA pipelines while showing its notable superiority in computational efficiency.

**Strengths:**

1.	The main contribution of this paper is the proposed event representation, which considers a balanced splitting of event stream and also leverages a spiking neuron-based model for encoding temporal information. Together with the specific efforts on further network elements, real-time performance is achieved with comparable accuracy than SoTA solutions.

2.	The paper is relatively easy to follow. Most of the claimed contributions are justified by comprehensive experiments.

The authors put significant efforts into optimizing the model for efficiency and achieving higher FPS by reducing model size and utilizing a more efficient representation network. Their analysis of energy and computational efficiency is thorough and comprehensive.

**Weaknesses:**

1.	One theoretical flaw of this paper is regarding the so called “optimally splitting of event stream”. Although it is endorsed via establishing an intuitive correlation between the probabilistic characteristics of event data and intensity images, it is still a heuristic setup that tries to preserve the event density to some extent. In fact, it is a GLOBAL threshold that cannot handle the various local dynamics of event activity [A] caused by different parallaxes (e.g., events stimulated by far and close structures when the camera undergoes pure translation). A more reasonable design is an adaptive splitting scheme that considers local dynamics of event activity.

2.	The proposed LIF model for encoding temporal information seems novel to me, and its advancement compared to other alternatives (e.g., LeakyReLu) is shown in the ablation study. However, readers may be curious about, from a theoretical perspective, the reason that LIF is used rather than other spiking neuron models.   More theoretical analyses are needed.

3.	Some clarifications (details) are needed. For example, the definition of “standard deviations” and “event densities” for each event grouping are better to be detailed in Sec. 3.1 with more theoretical analyses (see next).   The event stream pre-processing stage is the only difference to the existing SE-CFF and EI-Stereo pipelines, where "optimally splitting the event stream" is presented as a key contribution of the paper.  However, Section 3.1 offers empirical observations, lacking theoretical support or analysis. The rationale behind selecting a delta_t that achieves the closest standard deviation between event groupings and RGB images is unclear. There is no concrete ablation study on this point.


Reference [A] U. M. Nunes, R. Benosman, and S.-H. Ieng, “Adaptive global decay process for event cameras,” in IEEE Conf. Comput. Vis. Pattern Recog. (CVPR), 2023, pp. 9771–9780.

**Questions:**

The “skip connections” applied in Sec. 3.4 is not clear in the event domain; In line 318-320, how many event groups are passed to the representation network? Does each group return a unique representation, or N groups lead to only one representation ?

Please also respond to or comment on each of the weaknesses mentioned above.

Typo:  Line 215, a comma should be used after “LIF neuron model”.

---

### Official Review · Reviewer_FhaF · 2024-11-03

**Soundness:** 3
**Presentation:** 3
**Contribution:** 3
**Rating:** 5
**Confidence:** 4

**Summary:**

This paper presents a method for stereo depth estimation from paired intensity images and events from an event camera. The method is modified from existing standard architectures, consisting of feature extraction, cost volume computation, deformable aggregation and multi-level disparity prediction. The main contributions of this work are:
- A (mild) statistical study of the time window size to use for the events.
- The reduction of the convolutional layers in the feature encoder to 1x1 (effectively MLPs), and replacing the activations with spiking leaky integrate and fire neurons.
- A reduction in the size of the previous models (number of layers) and the introduction of skip connections to improve quality.
- Extensive experiments on the MVSEC and DSEC datasets demonstrating state of the art performance in both stereo depth quality and latency.

**Strengths:**

Overall the paper presents a mostly principled approach towards improving existing stereo intensity-event models. The authors identify several areas to improve (time window, activations, model size), and propose sensible improvements that move the needle for existing established public datasets. Overall, the paper is well structured and written, and the motivation and method is clear to the reader. Extensive ablations are also provided demonstrating the efficacy and efficiency of the proposed method, which provides a new SOTA for intensity-event stereo depth estimation.

**Weaknesses:**

Overall, this paper seems like a study of previous methods and how various parameters can be tuned to make them more efficient (there are novelties in the use of the leaky integrate and fire activations, but this does not seem to be the main focus of the paper). Given the experiments, it's fairly clear that the authors have found a novel configuration that outperforms the prior works. However, the usefulness of such studies is often due to the scale of the ablations provided, where readers can understand the impact of tuning each hyperparameter over a range.

The main weakness I see in this paper is that it is primarily prescriptive in the hyperparameter choices, with few results over hyperparameter sweeps provided. For example, in Section 3.2, the authors state that they replace the feature encoder with 2 convolutional layers with 1x1 kernel size. While this does improve latency, it's not clear that how this configuration was chosen, or what the impact would be of changing either of these parameters. This means that the results are not particularly useful outside of the configuration proposed by the authors, and limits the amount of information gleaned from the work (e.g. what is the performance-latency upper bound if readers have looser latency requirements). Section 3.3 is similar in that it reports only results for the proposed halved number of layers, without intuitions or results detailing how these results were achieved.

To me this weakness seems significant as, while a new SOTA method is useful, the search space for such hyperparameters is incredibly large, and I'm wary of the value of papers introducing new data points along the performance upper bound without an investigation into how the SOTA results are achieved, or what neighboring points on the performance-latency curve look like.

Additional comments:
The use of convolutional layers in Section 3.2 is a little misleading as the kernel size is reduced to 1x1. In this case, is the model equivalent to a pair of MLPs? By reducing the kernel size to 1x1, the behavior of the layers has significantly changed (no interactions between neighboring pixels).

I would suggest that the architectural overview in Section 4.1 be moved to the start of the methods section. For readers not familiar with existing event based pipelines, this provides a good high level overview of the method and comparison to existing works before delving deep into the contributions.

**Questions:**

It would be helpful for the authors to provide more information about how they got to the final configuration proposed in the paper, and whether it seems to be a locally-optimal one. However, it seems challenging to me to incorporate the set of studies I mention above without a significant rework of the paper.

---

### Official Review · Reviewer_sMxb · 2024-11-04

**Soundness:** 2
**Presentation:** 3
**Contribution:** 2
**Rating:** 3
**Confidence:** 4

**Summary:**

The paper presents ROSE, a framework designed for real-time stereo depth estimation using event cameras, addressing the computational overhead that hampers existing methods. ROSE aims to reduce model size and enhance speed while maintaining accuracy by leveraging lightweight event representation networks and optimizing the stereo matching process. The framework achieves impressive frame rates on the DSEC and MVSEC benchmarks.

**Strengths:**

Strengths:

1.ROSE significantly boosts processing speed, achieving frame rates of 32.2 FPS on DSEC and 66.9 FPS on MVSEC, which is a notable improvement over existing methods.

2.By releasing code and models, the authors promote further research and advancements in stereo depth estimation.

**Weaknesses:**

Weaknesses:

1.The paper reduces the model size by replacing the original complex event aggregation module with CNN+LIF. But apart from that, the paper only follows existing pipelines from previous works, lacking innovation in lightweight design.

2.As stated in the paper, the event aggregation module is very important for processing non-uniform density event inputs. But the paper can use a simple CNN+LIF with 1000 times reduced parameters to replace it. Is there anything unique about this module? Or is it more crucial to analyze and introduce priors about event distribution of datasets?

3.The authors only compare their method to the complex representation based on the U-Net structure from Nam et al. (2022), likely due to the limited number of works in this area. However, in the context of event-based models, there are many simpler representations available, such as voxel, event frame, and time surface. Numerous state-of-the-art algorithms for related tasks are built on these simpler representations, which typically involve negligible computational overhead. As a result, the current conclusions may lack persuasive strength.

4.The overall presentation feels more like a technical report than a comprehensive research paper. There is limited theoretical analysis or understanding of the specific tasks. The focus on tuning parameters across a small set of algorithms and datasets detracts from a deeper exploration of the method’s implications.

**Questions:**

Questions:
What unique advantages does the simplified CNN+LIF aggregation module offer compared to traditional methods, especially in scenarios with non-uniform event density?

Can the authors provide more insight into the theoretical foundations of their method, particularly regarding the selection of parameters and their impact on performance across varying conditions?

---

### Official Review · Reviewer_G5xj · 2024-11-04

**Soundness:** 3
**Presentation:** 3
**Contribution:** 3
**Rating:** 6
**Confidence:** 4

**Summary:**

This paper proposed a simple but effective method for event representation in the task of event-image stereo. It reduces the network size and computational cost that achieves real-time stereo estimation. It also analyzed the performance of binning different length of event streams in different datasets.

**Strengths:**

•	This paper introduces a lightweight network architecture designed to minimize both model size and computational cost. This efficiency is particularly valuable for applications requiring real-time processing on resource-limited devices, where maintaining a balance between performance and resource use is critical.

•	On benchmark event-stereo datasets, the proposed method achieves higher frames per second (FPS) without sacrificing accuracy. This capability positions the model as a strong candidate for deployment on edge devices, an important feature for embodied AI applications where rapid decision-making and low latency are essential.

•	The paper provides a thorough analysis of various approaches for splitting event streams, enabling an optimal event grouping strategy selection. This analysis is a valuable contribution, as it guides researchers and practitioners in choosing effective configurations based on specific task requirements, thereby enhancing adaptability and performance in different settings.

**Weaknesses:**

•	The primary contribution of this work appears limited to modifying the event representation used in existing networks, such as SE-CFF and EI-Stereo.

•	In selecting the temporal window parameter, $\delta t$, a static approach is used, which may limit the method’s adaptability across diverse scenarios. It would be beneficial to adopt a dynamic $\delta t$ adjustment strategy that can respond to environmental conditions and activity levels. For instance, as illustrated in Figure 7, scenes with varying lighting conditions (bright vs. low light) or motion speeds (high-speed vs. low-speed) would ideally require different $\delta t$ values to account for changes in event-triggering efficiency or frequency. Incorporating an adaptive $\delta t$ mechanism could enhance robustness across a wider range of scenarios, especially in real-world applications where conditions vary frequently.

**Questions:**

N/A

---

### Official Review · Reviewer_VsAD · 2024-11-04

**Soundness:** 2
**Presentation:** 3
**Contribution:** 2
**Rating:** 5
**Confidence:** 3

**Summary:**

This paper introduces a real-time framework for efficient depth estimation based on events and intensity images. The proposed method incorporates lightweight event representation and optimizes stereo-matching to reduce model size and computational load without compromising accuracy.

**Strengths:**

1. The visualization results show that the spiking neuron-based event representation proposed in this paper can effectively avoid blur in the long stack while preserving the structural data from the event data information, which is useful for estimating scene depth based on events.
2. The proposed network optimization strategy effectively reduces the complexity of the network, and the experimental results show that the proposed algorithm can achieve real-time depth estimation.

**Weaknesses:**

1. The authors conclude that the spiking neuron-based event representation proposed in this paper is one of the core contributions of this paper. The advantages of this representation over long stack and short stack representation on DSEC and MVSEC datasets are verified by experiments. However, the paper does not seem to introduce the implementation details and theoretical basis of this event representation in detail, that is, why this event representation can realize both blur-free and structure information extraction? If only two convolutional layers are used or existing Leaky integration-and-fire (LIF) spiking neurons are introduced, the contribution of this event representation is weak.
2. The algorithms compared in this paper were mainly published in 2022, and it is suggested to increase the performance comparison with the latest algorithms, such as "Temporal Event Stereo via Joint Learning with Stereoscopic Flow" published in ECCV2024. The code of the revised paper has been made public.
3. The "real-time" mentioned in the paper seems to refer to the increase in the frame rate of the output depth video, rather than the real-time output depth results while the camera is collecting data, whether this can be considered "real-time" is still to be further discussed.

**Questions:**

Please refer to the weaknesses.

---

### Note · Authors · 2024-11-13

**Comment:**

Based on the reviews, we need to update the draft significantly for a future conference. So, withdraw it.

**Withdrawal Confirmation:**

I have read and agree with the venue's withdrawal policy on behalf of myself and my co-authors.